# Activated Human Adipose Tissue Transplantation Promotes Sensorimotor Recovery after Acute Spinal Cord Contusion in Rats

**DOI:** 10.3390/cells13020182

**Published:** 2024-01-17

**Authors:** Maxime Bonnet, Céline Ertlen, Mostafa Seblani, Jean-Michel Brezun, Thelma Coyle, Cristina Cereda, Gianvincenzo Zuccotti, Mattia Colli, Christophe Desouches, Patrick Decherchi, Stephana Carelli, Tanguy Marqueste

**Affiliations:** 1Aix Marseille Univ, CNRS, ISM, UMR 7287, Institut des Sciences du Mouvement: Etienne-Jules MAREY, Equipe «Plasticité des Systèmes Nerveux et Musculaire» (PSNM), Parc Scientifique et Technologique de Luminy, CC910-163, Avenue de Luminy, CEDEX 09, F-13288 Marseille, Francejean-michel.brezun@univ-amu.fr (J.-M.B.); patrick.decherchi@univ-amu.fr (P.D.); 2Center of Functional Genomics and Rare Diseases, Department of Paediatrics, Buzzi Children’s Hospital, Via Ludovico Castelvetro 32, 20154 Milano, Italy; 3Pediatric Clinical Research Center «Romeo ed Enrica Invernizzi», Department of Biomedical and Clinical Sciences, University of Milano (UNIMI), Via G.B. Grassi 74, 20157 Milan, Italy; gianvincenzo.zuccotti@unimi.it; 4Department of Paediatrics, Buzzi Children’s Hospital, Via Ludovico Castelvetro 32, 20154 Milano, Italy; 5Podgora7 Clinic, Via Podgora 7, 20122 Milano, Italy; 6Clinique Phénicia—CD Esthétique, 5 Boulevard Notre Dame, F-13006 Marseille, France

**Keywords:** fat, reflex, inflammation, sensorimotor, paraplegia, neuroprotection, injury

## Abstract

Traumatic spinal cord injuries (SCIs) often result in sensory, motor, and vegetative function loss below the injury site. Although preclinical results have been promising, significant solutions for SCI patients have not been achieved through translating repair strategies to clinical trials. In this study, we investigated the effective potential of mechanically activated lipoaspirated adipose tissue when transplanted into the epicenter of a thoracic spinal contusion. Male Sprague Dawley rats were divided into three experimental groups: SHAM (uninjured and untreated), NaCl (spinal cord contusion with NaCl application), and AF (spinal cord contusion with transplanted activated human fat). Pro-inflammatory cytokines (IL-1β, IL-6, TNF-α) were measured to assess endogenous inflammation levels 14 days after injury. Sensorimotor recovery was monitored weekly for 12 weeks, and gait and electrophysiological analyses were performed at the end of this observational period. The results indicated that AF reduced endogenous inflammation post-SCI and there was a significant improvement in sensorimotor recovery. Moreover, activated adipose tissue also reinstated the segmental sensorimotor loop and the communication between supra- and sub-lesional spinal cord regions. This investigation highlights the efficacy of activated adipose tissue grafting in acute SCI, suggesting it is a promising therapeutic approach for spinal cord repair after traumatic contusion in humans.

## 1. Introduction

From a societal point of view, the incidence of spinal cord injury (SCI) exhibits regional and international variability, reflecting the influence of diverse factors such as the expanding scope of human activities. In 2013, the World Health Organization reported a staggering 500,000 new cases of SCI annually worldwide. A comprehensive meta-analysis encompassing studies from 23 developing countries revealed an overall incidence rate of 22.55 cases per million inhabitants per year [1].

Individuals with spinal cord injuries have a significantly reduced life expectancy compared to healthy individuals [2]. This is due to an increased probability of developing secondary disorders such as respiratory, cardiovascular, and digestive diseases, cancer, urinary dysfunction, and sepsis [2,3,4,5]. In addition to the profound physical and emotional toll, SCI imposes a substantial financial burden, encompassing hospital treatment costs and implications for both patients and their families [6,7]. The costs associated with SCI can be categorized into three phases: immediate medical intervention, direct costs, and indirect costs. In the United States, immediate medical interventions required to treat the injury and stabilize the patient’s condition range from USD 92,220 to USD 337,400 [8]. Annual direct costs encompass expenses related to the long-term management of SCI, including regular medical care and rehabilitation treatments, ranging from USD 27,437 to USD 69,469 [9]. Additionally, annual indirect costs associated with managing and adapting to life with SCI fall within the range of USD 35,246 to USD 100,838 [10].

From a pathophysiological point of view, SCI is a severe condition that compromises the sensory, motor, and autonomic functions of the body, significantly affecting patients’ overall quality of life, including their psychological and psychosocial well-being [6,7,11]. Functional studies reported changes in the connectivity of cortical sensorimotor regions after SCI [12]. MRI investigations have recently shown increased sensorimotor network connectivity in complete SCI, indicating that the connectivity had changed because of post-injury events [13]. Indeed, the pathogenesis of SCI unfolds through a primary mechanical trauma followed by a secondary cascade of cellular and molecular events, contributing to the propagation of tissue damage. This intricate sequence of events culminates in the impairment or malfunction of crucial neural relay circuits that govern respiratory function, motor skills, bladder control, autonomic regulation, and may also induce sensory dysfunctions [10]. In rodents, spinal cord contusion disrupts various spinal tracts situated in both the white matter and central gray matter, resulting in alterations to the neural circuitry. Studies have demonstrated that the disruption of these circuits strongly correlates with locomotor and neuron impairments, particularly in the spared white matter (WM). The WM houses critical tracts, such as the reticulospinal, spinothalamic, posterior column-medial lemniscus, and propriospinal tracts, and the degree of preservation in these regions significantly influences motor and sensory outcomes [14,15]. These events which follow the SCI have been highly investigated and reviewed [16]. Briefly, the primary injury, caused by mechanical impact, triggers a cascade of complex secondary processes, including inflammation, excitotoxicity, oxidative stress, demyelination, and cell death. These contribute to further spinal cord degeneration, consolidation of functional deficits, and impairment of the recovery [16]. Thus, a more precise understanding of SCI resultant lesional environment variability could clarify differences in endogenous plasticity and repair responses [17].

Even if recent clinical trials have demonstrated that spinal neurostimulation allowed some patients to regain the ability to move [18,19] and transplantation of olfactory ensheathing cells (OECs) allowed patients with partial SCI to regain motor function [20], all clinical strategies proposed to limit the progression of the damage and to improve outcomes in SCI (including high-dose administration of methylprednisolone, surgical interventions for vertebra stabilization and tissue decompression, and rehabilitative care), showed that the functional preservation and recovery in patients with complete SCI still remains unsatisfactory [21,22,23,24,25,26].

Basic research on animal models has also extensively explored various strategies aimed at protecting damaged spinal cord, reducing secondary injury, inhibiting glial scar formation, and promoting axonal regeneration [27,28]. Among these strategies, cellular therapies have emerged as a promising approach for repairing SCI through transplantation of various cell types. Indeed, neural stem and progenitor cells (NPCs), oligodendrocyte progenitor cells (OPCs), mesenchymal stem cells (MSCs), Schwann cells, OECs, and emerging stem cell scaffolds have all shown a potential for inducing functional improvement after SCI in animal studies [29,30,31,32,33,34]. The latter cell types were investigated for their neurotrophic and neuroprotective abilities, replacement of dead cells, immunomodulation, axon sprouting and/or regeneration, and myelinization [29,31,35]. The diverse mechanisms of action exhibited by these cell types makes them potential candidates for cell therapy in SCI treatment. However, despite promising results in preclinical studies, the translation of these strategies to clinical trials has not led to significant solutions for SCI-affected patients [21,23,26,30,36]. There remains a critical need to explore novel therapeutic approaches able to effectively address the complex pathophysiology of SCI and promote functional recovery in patients.

The use of adipose tissue as a therapeutic approach for SCI looks very promising. Contrary to what has long been thought, adipose tissue does not only contain mature adipocytes, but it is a true endocrine organ, secreting many protective adipokines such as leptin, angiotensinogen, resistin, visfatin, and adiponectin [37,38,39,40]. It is also an abundant source of cells such as preadipocytes, fibroblasts, pericytes, macrophages, T-cells, endothelial progenitor, and mesenchymal stem cells (MSCs). These MSCs are also called adipose-derived stem cells (ADSCs); they exhibit neuronal markers and have high regenerative potential due to their ability to differentiate into other tissue types [33,41,42,43,44,45]. Compared to other sources of MSCs, subcutaneous adipose tissue can be easily accessed with minimal patient discomfort, and the isolation procedures for obtaining ADSCs are easier [32,46,47]. Indeed, in experimental rat models of SCI, ADSCs were found to promote functional recovery and neovascularization [48,49]. However, although clinical trials using autologous transplantation of ADSCs to treat acute SCI have shown some improvement in terms of motor function, bladder function, and daily living, recovery has remained insufficient [50].

With respect to functional recovery, recent studies have shown that mechanical forces applied to the adipose tissue can improve its protective and regenerative properties. Specifically, these effects were obtained by upregulating the expression of anti-inflammatory cytokines, like TNF-α-stimulated gene-6 (TSG-6), and favoring the expression of endogenous neural markers in the treated spinal cord [33,43]. This mechanical activation triggers mechano-transduction pathways which involve ion channels, growth factors, cytoskeletal filaments, focal adhesions, and tyrosine kinases [33,51,52,53]. These pathways lead to signals which communicate messages to the nucleus to modulate gene expression [33,43,54,55]. As the need for prolonged in vitro cell culture is eliminated, this approach provides a more convenient and faster procedure, constituting a promising approach to providing MSCs for SCI treatment [33].

In this study, we aimed to investigate the regenerative potential of mechanically activated lipoaspirated human adipose tissue transplanted into the epicenter of lesioned spinal cord in a reproducible and widely used rat experimental model [56]. We investigated the benefit of using mechanically activated fat in an experimental model of acute spinal cord contusion in rats and focused on sensorimotor recovery, the neural networks involved, and supraspinal modulation. The transplantation was performed immediately after the contusion, and the treated rats were compared to control animals without SCI (SHAM group) and to lesioned animals which received a saline solution (NaCl group). To assess sensorimotor recovery, we conducted a weekly evaluation using the BBB locomotor rating scale [57] and ladder rung climbing test [58,59] for 12 weeks, both before (PRE-) and after the lesion (W1 to W12). Additionally, a separate group of animals was sacrificed two weeks post-injury to evaluate the potential anti-inflammatory effect exerted by the fat transplant [60]. At the end of the sensorimotor evaluation period, gait analysis was performed. Finally, to assess the functioning of the sensorimotor loops under the lesion and the control of supraspinal levels, animals were anesthetized for electrophysiological recording to investigate H-reflex and ventilatory adjustment to muscle fatigue [61].

## 2. Materials and Methods

### 2.1. Animals

The experiments were performed on 33 adult male Sprague Dawley rats, weighing between 180 and 200 g (Centre d’Élevage Roger JANVIER^®^, Le Genest Saint Isle, France), hosted two per cage in smooth-bottomed plastic cages in a laboratory animal house maintained on a 12:12 h light/dark photoperiod and at 22 °C. Drinking water and rat chow (Safe^®^, Augy, France) were available ad libitum. The animals were housed in the animal facility for 2 weeks before the initiation of the experiment. This habituation period allowed the decrease of inter-individual differences in order to reach optimal performance. At the end of this period, PRE-values for each behavioral test (BBB test, ladder-climbing test, and gait analysis) were recorded.

### 2.2. Ethical Considerations

The experiments were conducted according to the French legislation (decrees and orders N°2013-118 of 1 February 2013, JORF n°0032) concerning animal care guidelines on animal experimentation and following the approval by the Animal Care Committees of Aix-Marseille Université (AMU) and Centre National de la Recherche Scientifique (CNRS). The authorization number granted by the French Ministry of Higher Education, Research, and Innovation (MESRI) is APAFIS#32782. All persons are licensed to conduct live animal experiments and all experimental rooms have a national authorization to accommodate animals (license n°B13.013.06). Furthermore, the experiments were performed in accordance with the recommendations provided in the Guide for Care and Use of Laboratory Animals (U.S. Department of Health and Human Services, National Institutes of Health), with the directives 86/609/EEC and 010/63/EU of the European Parliament and of the Council of 24 November 1986 and of 22 September 2010, respectively, and with the ARRIVE (Animal Research: Reporting of In Vivo Experiments) guidelines.

After surgery, the animals were placed under a heat lamp until their thermoregulation was restored. The health status of the animals was monitored daily, and any animals showing signs of distress such as vocalization, lethargy, hyperactivity, significant weight loss (15 to 20% of initial weight), and self-mutilation behavior were euthanized. All animals operated on received subcutaneous injections (3 mL) of glucose-enriched saline solution to replace the fluid lost during the surgical procedure two to three times per day until they were able to drink and eat on their own. Buprenorphine (0.03 mg/kg, 0.3 mg/mL, Bruprécare^®^ Multi-dose, Axience Santé Animale SAS, Pantin, France) was administered subcutaneously daily for 5 days, and amoxicillin trihydrate (Clamoxyl^®^ L.A, 800 mg, Zoétis France S.A.S, Malakof, France) was diluted (13–20 mg/kg) in drinking water for 1 week to prevent any bacterial infections. If necessary, bladder expression was manually performed at least twice a day. Postoperative nursing care also included visual inspection for skin irritation or pressure ulcers, followed by cleansing the hindquarters with soap and water and rapid drying of the fur with a towel.

### 2.3. Fat Collection, Purification, and Mechanical Activation

Liposuction aspirates from peri-umbilical subcutaneous adipose tissue were obtained from 3 healthy young adult women volunteers (27.67 ± 1.33 years old) undergoing elective plastic surgery procedures. The protocol was reviewed and approved by the Assistance Publique des Hôpitaux de Marseille (AP-HP) before the study. The liposuction procedure was performed after the consent obtained from the informed donors by signing a specific informed consent form.

Local anesthesia was achieved with an infiltration of lidocaine hydrochloride (Xylocaine^®^ 20 mg/mL, Aspen Pharmacare Canada Inc., Oakville, ON, Canada). As part of the procedure, a solution of saline and epinephrine (2 μg/mL) was infused into the adipose compartment to minimize blood loss and reduce infiltration by peripheral blood cells.

To activate the lipoaspirated adipose tissue, a method described in previously published research was followed [33,43]. Briefly, 10–15 mL of lipoaspirated adipose tissue was mechanically activated by orbital shaking at a speed of 97× *g* (N·kg^−1^) for a period of 7 min at room temperature. The fat was activated 30–60 min before transplantation. The resulting product was referred to as mechanically activated lipoaspirate or activated fat (AF).

To demonstrate that the procedure of activation of lipospirated adipose tissue does not significantly alter the vitality of the tissue, we isolated human adipose tissue cells using a procedure previously described [19]. Briefly, 1 cm^3^ of AF was plated in a dish (10 cm diameter) and the growth medium was added without covering the tissue and allowing its adhesion to the bottom surface of the dish. After a few days in culture at 37 °C and 5% CO_2_, the cells started to grow and after 15–21 days the cells reached 50% confluence. The obtained cells were expanded and characterized for the expression of endothelial, hematopoietic, and mesenchymal markers.

### 2.4. Experimental Groups and Design of the Study

Following a two-week period of acclimatization, during which the animals underwent familiarization sessions lasting one hour per day, three days per week, involving the open-field test (BBB test), inclined ladder test (ladder-climbing test), and walking corridor (gait analysis), reference values (PRE-) were measured for each behavioral test. Subsequently, the animals were randomly allocated to the three experimental groups: (1) SHAM (n = 11), where a surgery with a laminectomy without spinal lesion was performed; (2) NaCl (n = 11), where a thoracic T10 contusion was induced (W0), followed by immediate application of NaCl on the lesion area; and (3) AF (n = 11), where 150 µL of mechanically activated human adipose tissue was applied immediately after contusion to the exposed dura mater of the spinal cord around the lesion area.

Two weeks (W2) after the injury, a subset of these animals from each group (n = 5 per group) was sacrificed to assess endogenous inflammation at the site of the spinal lesion. The remaining animals (n = 18; 6 animals in each group) were subjected to weekly assessments of sensory and motor recovery in the hindlimbs. Baseline measurements were taken one week prior to the injury (PRE-), and evaluations were continued from one week post-injury (W1) up to twelve weeks (W12) thereafter. The recovery progress was monitored using behavioral tests, namely, the BBB (Basso–Beattie–Bresnahan) and ladder-climbing tests. At the end of the twelve-week period (W12), a gait analysis test was performed, and sensorimotor loops were evaluated through electrophysiological recordings of the M-wave and H-reflex below the lesion site. Additionally, after inducing muscle fatigue, the medullary ventilatory center was evoked via the metabosensitive pathway. The chronological order of our protocol is schematically shown in Figure 1.

### 2.5. Surgical Procedure for the SCI

As previously described [56], the surgical procedure for exposing and damaging the spinal cord consisted firstly of anesthetizing the rat with an intraperitoneal injection of a mixture containing ketamine hydrochloride (75 mg/kg, 100 mg/mL, Ketamidor^®^, Axience S.A.S., Pantin, France) and medetomidine hydrochloride (0.5 mg/kg, 0.85 mg/mL, Domitor^®^, Vetoquinol, S.A., Lure, France), along with an injection of buprenorphine (0.03 mg/kg, 0.3 mg/mL, Bruprécare^®^ Multi-dose, Axience, Axience S.A.S., Pantin, France). The animal was placed in a ventral decubitus position, and the back was shaved and disinfected with a 10% povidone-iodine solution (Vétédine^®^ Solution, Vetoquinol S.A, Lure, France).

A midline dorsal incision was made over the T6-T13 spinous processes, and the superficial muscles were retracted using retractors to expose the thoracic vertebrae. A dorsal laminectomy was performed at the T10 level to expose the spinal cord without compromising its integrity. The dura mater was left intact, and stabilization clamps were placed at the posterior processes of vertebrae T9 and T11 to support the vertebral column during traumatism.

Spinal cord contusion was induced at the 10th thoracic vertebra (T10) level using an IH-0400 weight-drop impactor (Infinite Horizon Impactor^®^, Precisions Systems and Instrumentation, LLC, Lexington, KY, USA). The impact strength was set at 200 kdyn, considered by previous studies to be a moderate [15] or a severe [62] lesion for animals of equivalent weight.

Following SCI, the animals in the NaCl and AF groups received either NaCl or an AF application, respectively, to the exposed dura mater at the lesion site (Figure 2). The muscles and the skin were then closed using synthetic (polyglactine 910 thread coated with polyglactine 370 and calcium stearate) and absorbable and braided sutures (Vicryl^®^ 3-0) for muscle and skin, respectively (Ethicon^®^, Issy-les-Moulineaux, France).

### 2.6. Endogenous Inflammation

Two weeks (W2) after the SCI and transplantation, the endogenous inflammatory response at the lesion site was assessed as previously described [56]. Briefly, after euthanasia, a segment of the spinal cord extending 5 mm rostral and caudal to the injury site was carefully harvested and immediately immersed in isopentane. The samples were then stored at −80 °C until further analysis.

Once all the collected samples were ready, the samples were shortened by around 1 mm rostral and caudal to the lesion. They were then individually homogenized in 1 mL of phosphate buffer saline (PBS) using a handheld homogenizer (Ika Ultra Turrax^®^ disperser, Fisher Scientific SAS, Illkirch, France) equipped with plastic pestle tips which facilitate tissue homogenization through vibrating motions. The resulting mixtures were subsequently centrifuged (Sigma 2-16 PK Centrifuge, Fisher Scientific SAS, Illkirch, France) for 12 min at 12,000× *g* and 4 °C. A fraction of the supernatant (50 µL) containing soluble proteins was used for the evaluation of pro-inflammatory cytokines.

The concentrations of pro-inflammatory cytokines (interleukin-1β (IL-1β), interleukin-6 (IL-6), and tumor necrosis factor-alpha (TNF-α)) were measured using enzyme-linked immunosorbent assay (ELISA) kits that included specific antibodies (RAB0272, RAB0311, and RAB0480; Sigma Aldrich^®^, Saint-Quentin Fallavier, France), following the manufacturer’s instructions. All samples were run in duplicate, and the absorbance was read at a wavelength of 450 nm using a microplate reader (Multiskan^®^ Microplate Photometer, Thermo Fisher Scientific, Life Technologies SAS, Courtaboeuf, France). Concentrations were determined by referencing a standard curve and considering the weight of the tissue prior to homogenization. Consequently, the levels of cytokines were expressed as picograms per gram (pg/g) of spinal cord tissue.

### 2.7. Evaluation of Hindlimb Recovery

Prior to the surgery, baseline reference values (PRE-) for each behavioral test were measured. Subsequently, for a period of 12 weeks, starting from one week after the injury (W1) and until the twelfth week (W12), the BBB and ladder-climbing tests were conducted once a week. At W12, gait analysis data were collected. Throughout these assessments, the experimenters involved in the data collection were blinded to the treatment group, ensuring unbiased evaluations.

BBB test. The locomotor functions were assessed using the Basso–Beattie–Bresnahan (BBB) test [57]. The test involved placing the animals in a circular Plexiglas^®^ enclosure arena with an anti-slip floor, measuring 95 cm in diameter and 40 cm in wall height. The animals were allowed to freely walk in the open-field environment, and their locomotion was recorded using a camcorder (GoPro Hero11^®^, GoPro, Inc., San Mateo, CA, USA). The duration of each session was 4 min. The analysis of locomotor score was performed later using the BBB scale, which consists of 22 levels representing different stages of locomotion, ranging from complete paralysis (level 0) to normal locomotion (level 21). For each rat, the scores of the two hindlimbs were averaged to obtain a single score for each test session.

Ladder-climbing test. Fine sensorimotor coordination was tested during climbing of a 45°-inclined ladder (100 mm × 1500 mm). This test does not require conditioning and evaluates the capacity of the animals to correctly place their paws on round metal rungs (4 mm in diameter) spaced at equal intervals of 20 mm, with 150 mm high side walls [56,59,61,63]. The animals were placed at the bottom of the ladder, and their climbing behavior was video-recorded from a position below the ladder, providing a ventral view of both hindlimbs using a camcorder (GoPro Hero11^®^). At the top of the ladder, the animals had access to a dark box. The video recordings were carefully analyzed using slow-motion playback. The placement of the hind paws on the ladder rungs was scored as follows: 0: hindlimb hanging in front of or behind the rungs without supporting climbing, 1: hindlimb used to support climbing but hind paw not placed correctly on the rung, and 2: hind paw correctly placed on the rung, maintaining the position while the trunk and the contralateral limb moved upward. The scores obtained on each side were averaged, and a climbing score ranging from 0 (no successful grip with the hind paws) to 20 (climbing 20 rungs of the ladder without faults, i.e., 10 grips with each hind paw) was calculated. To evaluate the progress of the animals over time, the mean climbing score ratio obtained at each session was expressed as a percentage of the mean ratio obtained at week 0 (PRE-).

Gait analysis. Motor function and coordination were quantitatively assessed using a custom-made gait analysis system. The system consisted of a long glass walking plate placed in a corridor (110 cm long, 8 cm wide, with opaque walls 15 cm tall), a fluorescent light directed onto the glass plate, and a camcorder (GoPro Hero11^®^) positioned beneath the glass plate. The recordings were conducted in a dark environment, with the footprints of the animal generating a downward reflection of the light which was recorded on the video. To obtain data, the animal needed to achieve at least a weight-supported stepping (BBB score of 9 or higher) [64,65]. The animal performed unforced and uninterrupted crossings on the glass walkway at least three times. For each run, average values were calculated for further analysis of the parameters.

Gait analysis parameters were collected for each animal. The first and last steps, which correspond to the run-up and slow-down phases, were excluded [66]. Their five best consecutive steps were chosen based on the limit length of the device corridor. This method was used to best represent the walking performance. Data analysis was performed using a customized function developed in the MATLAB R2022b environment (The MathWorks, Inc., Natick, MA, USA). The function was based on the MouseWalker system [67], with the added feature that any incorrectly labeled footprints were manually corrected frame by frame.

The following parameters were studied to evaluate motor function and coordination during gait analysis:− Average speed: The speed of forward locomotion across the runway, measured in centimeters per second (cm/s).− Stride length: The distance between successive placements of the same paw, measured in centimeters (cm).− Stance phase: The duration of paw contact with the glass plate, measured in seconds (s).− Swing phase: The duration during which the paw was not in contact with the glass plate, measured in seconds (s).− Step cycle: The time between two successive placements of a single paw, calculated as the sum of the stance and swing phases, measured in seconds (s).− Phase dispersion: This parameter evaluated the synchrony of the initial contact between pairs of limbs. It indicates the timing between the first contacts of paw pairs (e.g., RF-LH) relative to the step cycle of the anchor paw. Phase dispersion is expressed as a percentage (%).− Step sequence patterns: The stepping patterns were categorized as normal when the animal sequentially placed its four paws in an alternating, cruciate, or rotating pattern.− Regulatory index (RI): This index measures the degree of coordination among the limbs and is expressed as a percentage (%).

### 2.8. Electrophysiological Recordings

At 12 weeks post-injury (W12), the animals underwent electrophysiological recordings after being prepared according to previously described procedures [56,61,68,69]. Briefly, animals were initially sedated with isoflurane (3–5% in oxygen, Isoflo^®^ 100%, Zoetis SAS, Malakoff, France). Then, to ensure deep anesthesia, a mixture containing ketamine hydrochloride (75 mg/kg, 100 mg/mL, Ketamidor^®^) and medetomidine hydrochloride (0.5 mg/kg, 0.85 mg/mL, Domitor^®^) was administered via an intraperitoneal injection. The peroneal nerves from both hindlimbs were carefully dissected free from the surrounding tissues. This step allowed for nerve stimulation during the electrophysiological recordings. The Tibialis anterior muscles from both hindlimbs were exposed to facilitate electromyographic (EMG) recording and stimulation.

Physiological reflexes. As previously described, ventilatory adjustments were measured through a cannula inserted into the trachea during and after the Tibialis anterior muscles’ stimulation [41,52,53,55,58,59]. The changes induced by electrically induced muscle fatigue (EIF) were expressed as a percentage (Δcycle/min (%)) of the mean cycles recorded two minutes before the muscle stimulation.

M- and H-waves. As previously described, the Tibialis anterior muscles’ M- and H-waves were recorded after electrical stimulation of the peroneal nerve [56,68,69,70,71,72,73]. The rate-dependent depression (RDD), also known as rate-sensitive depression (RSD), frequency-related depression (FRD), frequency-dependent depression (FDD), or low-frequency depression (LFD) was examined. For that, the Hmax and Mmax were recorded and the Hmax/Mmax ratio was calculated at different stimulation frequencies (1, 5, and 10 Hz) and compared to the Hmax/Mmax ratio obtained at the baseline frequency of 0.3 Hz. This analysis allowed for the assessment of the decrease in reflex magnitude relative to the repetition rate of stimulation.

### 2.9. Euthanasia

According to ethical recommendations, at the end of the electrophysiological recordings, animals were killed with an i.p. overdose of pentobarbital sodium (390 mg/kg, Euthasol^®^ Vet.) and the spinal cord was removed and stored at −80 °C for future analysis.

### 2.10. Statistical Analysis

Statistical analyses were performed using the SigmaStat–SigmaPlot 14.0 software environment for statistical computing (SigmaStat^®^, San Jose, CA, USA) and GraphPad Prism 8.0.2 for graphics (GraphPad Prism Software Inc., San Diego, CA, USA). The analysis of the ELISA data was performed using a one-way design experiment (groups factor), with a Kruskal–Wallis one-way analysis of variance, followed by a Dunn test. For two-way design experiments (groups factor × time or stimulation frequency factor) we used Scheirer–Ray–Hare, followed by a Dunn test, allowing a comparison of behavioral scores from all groups and over time and Hmax/Mmax ratios from all groups. Data were expressed as median ± SD and differences were considered to be significant when *p* < 0.05.

## 3. Results

### 3.1. Expression of Mesenchymal Markers

In vitro characterization of cell cultures from AF showed that expanded cells expressed endothelial, hematopoietic, and mesenchymal markers (Figure 3).

### 3.2. Animals

After SCI, all animals exhibited dramatic and bilateral hindlimb paralysis with no movement or only slight joint movements. Throughout the experimental period, the weight of the animals did not show any significant drop, indicating that their overall health and nutritional status were maintained. Furthermore, we did not observe any significant difference in terms of body weight among the different groups at W12. The mean weight of all rats was 280 ± 5 g at the beginning of the experiment and 660 ± 30 g, 648 ± 43 g, and 652 ± 36 g at W12 in the SHAM, NaCl, and AF groups, respectively.

### 3.3. Endogenous Inflammation

Measurement of IL-1β, IL-6, and TNF-α concentrations at the lesion site revealed differences in the inflammatory response among the groups two weeks after the SCI. Specifically, the SHAM and AF groups exhibited a lower level of pro-inflammatory cytokines compared to the NaCl group. Indeed, compared to the NaCl group, the concentrations of IL-1β, IL-6, and TNF-α were significantly lower (*p* < 0.001) in the SHAM group and the concentrations of IL-6 and TNF-α were significantly lower (*p* < 0.05) in the AF group. No difference was found between the SHAM and AF groups with respect to the concentrations of IL-6 and TNF-α. Only, the concentration of IL-1β was higher (*p* < 0.01) in the AF group compared to the SHAM group (Figure 4). 

### 3.4. Behavioral Tests

BBB test. The analysis of the BBB scores revealed a significant drop (*p* < 0.001) at W1 in all groups with spinal cord lesions (NaCl and AF) compared to pre-injury (PRE-) scores. Although the scores of these two groups remained lower than the SHAM group throughout the study, a slow recovery was observed during the following eleven weeks. Specifically, in the AF group the improvement in the BBB score was significantly appreciable already at W1 post-injury. Moreover, the improvement in the AF-treated group was continuous for the following observational period. At W12, the NaCl group achieved a score of 12 ± 0.52, indicating an intermediate stage characterized by intervals of uncoordinated stepping. Interestingly, the AF group showed a higher score of 16 ± 4.59, indicating a late stage characterized by consistent forelimb and hindlimb coordination with consistent weight support (Figure 5). 

Ladder-climbing test. The analysis of the ladder-climbing scores during the 12 weeks following the surgery revealed distinct patterns between the un-lesioned SHAM group and the two lesioned groups (NaCl and AF). The SHAM group demonstrated relatively stable scores, close to the maximal pre-surgery values, throughout the study period. In contrast, both lesioned groups showed a significant drop (*p* < 0.001) in scores at W1 post-surgery. Even though the scores in the two lesioned groups remained lower than the SHAM group over the 3-month period, interestingly, in the AF group, a fast recovery was observed from W3 to W6, indicating an improvement in climbing ability during this period. Furthermore, at W12, the score in the NaCl group (37.88 ± 8.10%) did not reach the score of the AF group (63.87 ± 32.20%) (Figure 6).

Gait analysis. At W12, the regulatory index was 100% in the SHAM and AF groups, and significantly (*p* < 0.01) reduced in the NaCl group (66.67 ± 2.23%). Moreover, animals from the NaCl (15.78 ± 3.27%) group showed a significant increase in phase dispersion compared to the SHAM (3.13 ± 1.02%, *p* < 0.01) and AF (3.43 ± 2.04%, *p* < 0.05) groups. No difference was found between the SHAM and AF groups (Figure 7A). Furthermore, there were no significant differences in the average crossing speeds on the glass plate between the three groups (SHAM: 41.61 ± 6.88 cm·s^−1^; NaCl: 21.14 ± 10.65 cm·s^−1^; AF: 39.92 ± 11.79 cm·s^−1^) (Figure 7B). However, the stride length was significantly decreased (*p* < 0.01) for the NaCl (12.60 ± 1.74 cm) group compared to the SHAM (15.92 ± 1.53 cm) and AF (14.77 ± 2.43 cm) groups. No difference was found between the SHAM and AF groups (Figure 7C). Finally, there was no difference between the three groups concerning the time-of-step cycle (SHAM: 0.38 ± 0.05 cm·s^−1^; NaCl: 0.34 ± 0.13 cm·s^−1^; AF: 0.33 ± 0.05 cm·s^−1^) (Figure 7D), the time-of-swing phase (SHAM: 0.14 ± 0.015 cm·s^−1^; NaCl: 0.12 ± 0.03 cm·s^−1^; AF: 0.13 ± 0.02 cm·s^−1^) (Figure 7E), and the time-of-stance phase (SHAM: 0.24 ± 0.04 cm·s^−1^; NaCl: 0.23 ± 0.13 cm·s^−1^; AF: 0.20 ± 0.04 cm·s^−1^) (Figure 7F).

Examination of the frequency of regular step patterns revealed a prevalence of the “alternate” step pattern across all groups (97.78% for the SHAM group, 84.44% for the AF group, and 70.59% for the NaCl group). The “cruciate” and “rotate” step patterns, typically employed by healthy animals, accounted for 2.22% and 0%, respectively, in the SHAM group, 2.22% and 13.33% in the AF group, and 17.65% and 11.77% in the NaCl group (Figure 7G).

### 3.5. Electrophysiological Recordings

Analysis of ventilatory adjustments during and after the activation of metabosensitive afferent fibers by repetitive stimulation of the Tibialis anterior muscle and under regional circulatory occlusion showed that there was a significant increase in ventilatory frequency (*p* < 0.001) only in the SHAM and AF groups. Additionally, data analysis revealed a significant difference between the NaCl group and the SHAM and AF groups during (*p* < 0.01) and after (*p* < 0.001) stimulation. Notably, the NaCl group exhibited an absence of ventilatory response. No significant difference was observed between the SHAM and AF groups (Figure 8).

The baseline Hmax/Mmax ratios measured at a stimulation frequency of 0.3 Hz were 0.36 ± 0.12, 0.32 ± 0.11, and 0.34 ± 0.12 for the SHAM, NaCl, and AF groups, respectively, and they showed no statistical difference. Analysis of the Hmax/Mmax ratio when increasing stimulation frequency revealed its decrease only in SHAM and AF groups. Indeed, in the SHAM group, the Hmax/Mmax ratio values at 1, 5, and 10 Hz were 97.26 ± 9.00%, 82.06 ± 15.71%, and 65.74 ± 19.88% of the ratio measured at the baseline stimulation, respectively. In the AF group, the corresponding values were 89.84 ± 9.62%, 63.57 ± 14.99%, and 49.68 ± 14.47%, respectively. In the NaCl group, the Hmax/Mmax ratio values at 1, 5, and 10 Hz were 95.89 ± 4.72%, 89.70 ± 16.81%, and 85.54 ± 12.19%, respectively. Data analysis revealed significant differences between the NaCl and AF groups at 1 Hz (*p* < 0.05), 5 Hz (*p* < 0.001), and 10 Hz (*p* < 0.001), and with the SHAM group at 5 Hz (*p* < 0.05) and 10 Hz (*p* < 0.01). No significant difference was found between the SHAM and AF groups regarding the depression of the H-reflex response at any of the tested frequencies (Figure 9).

## 4. Discussion

In this work, we investigated the efficacy of grafting mechanically activated human lipoaspirate on sensorimotor recovery after an acute spinal cord contusion in rats. We hypothesized that the activated fat, directly transplanted after the SCI at the lesion site, would reduce the neuroinflammation by reducing the amounts of the cytokines TNF-alpha and IL-6, two master regulators of chronic inflammation, and induce improvements in sensorimotor recovery.

Our results indicated that after a spinal cord thoracic contusion the inflammatory response was significantly reduced two weeks after the SCI in the treated animals (AF group) compared to the non-treated animals (NaCl group). In addition, the data showed locomotor recovery that was maintained over the 3-month study period. Finally, electrophysiological recordings twelve weeks after the trauma indicated a restoration of the sensorimotor loop and reconnection between supra- and sub-lesioned spinal cord.

These findings suggest that mechanically activated fat may provide neuroprotective and/or neuroregenerative factors which can increase tissue preservation following a secondary injury created by a series of biological and functional changes and/or regrowth of injured axons.

### 4.1. Activated Fat Reduced Neuroinflammation after SCI

Following SCI, the immediate consequence of trauma is a neuroinflammatory response at the injury site, characterized by local secretion of pro-inflammatory cytokines [74] and infiltration of peripheral immune cells such as macrophages, neutrophils, lymphocytes, leukocytes, and mast cells [66,75,76]. This reaction initiates within the first hours following the injury and persists for several weeks, with a peak of reactive astrocytes and macrophages around fourteen days post-injury [77]. These events, which are involved in the healing processes, can induce, if allowed to persist, tissue degeneration, further exacerbating the damage caused by the injury and limiting spontaneous regeneration [16].

Several studies have investigated the use of xenogeneic transplants of human ADSCs to repair the spinal cord after SCI [45,78,79,80]. These studies have shown that such grafts usually do not trigger immune responses. For instance, adipose tissue has the property of not expressing human leukocyte antigen II (HLA II), which is responsible for initiating the immune response [81,82]. Additionally, ADSCs secrete factor H, which strongly inhibits complement activation and prevents rejection of the fat graft [83]. In a previous study, the activated fat graft was used as a therapeutic approach for spinal cord repair after thoracic contusion in mice. In this study, the authors demonstrated the engraftment of adipose tissue at the cord injury site 35 days after contusion and transplantation [43]. Similarly, in our study we observed that the graft was still present around the lesion area fourteen days after the injury and transplantation (as reported previously in [43]). Consequently, the sustained presence of the graft may contribute to the observed modulation of inflammation.

Our study was, therefore, initially devoted to verifying that the graft of activated fat did not induce additional endogenous inflammation beyond that which develops spontaneously after SCI. For this reason, levels of the pro-inflammatory cytokines IL-1β, IL-6, and TNF-α were measured at the injury site at the 14-day time point. Our results showed a significant reduction in the concentrations of TNF-α and IL-6 in the AF group compared to the NaCl group, while no difference was observed between the AF and SHAM groups. Additionally, the concentrations of IL-1β were similar between the untreated injured group and the treated injured group. This means that activated fat treatment does not induce additional endogenous inflammation and helps in limiting the activation of a chronic response by counteracting the increase in IL-6 and TNF-α concentrations at the injury site.

The fat used in our study was obtained through liposuction of adipose tissue from young adult women. This adipose tissue contains elements with anti-inflammatory properties, such as adiponectin and ADSCs, which could explain the reduction in neuroinflammation observed in our study. Adiponectin is a hormone secreted by adipocytes naturally present in adipose tissue, known for its anti-inflammatory effects. Indeed, studies have shown that adiponectin can interrupt the activation of pro-inflammatory M1 macrophages and support the activation of anti-inflammatory M2 macrophages, thereby limiting the production of TNF-α and IL-6 cytokines while increasing the release of the anti-inflammatory interleukin IL-10 [84]. Other studies have revealed that the plasma adiponectin concentration is higher in women than in men [85] and it is able to cross the blood–brain barrier [86]. Leptin, an adipokine primarily secreted by white adipose tissue, was shown to cross the blood–spinal cord barrier (BSCB) and have anti-inflammatory effects following SCI. Fernández-Martos and al. demonstrated that leptin reduced the expression of inflammatory genes, such as IL-1β and nitric oxide synthase (iNOS), which are typically increased in the CNS after SCI in rats [87].

Additionally, Zhou et al. demonstrated, in a mouse model of SCI, that the graft of ADSCs at the site of the lesion was able to reduce the expression of anti-inflammatory cytokines IL-1β, IL-6, and TNF-α through the inactivation of the Jagged1/Notch signaling pathway [88].

In line with our findings, a previous study conducted by Carelli et al. highlighted the enhanced immunomodulatory properties of adipose tissue following mechanical activation [33]. They demonstrated that adipose-derived stem cells (ADSCs) from mechanically activated human adipose tissue exhibit robust anti-inflammatory properties. They inhibited the expression of the cytokines TNF-α and IL-1β and stimulated the expression of the inflammatory protein TSG6 (TNF-stimulated gene-6) when co-cultured with monocytes (THP1 cells) activated by lipopolysaccharide [33]. The increase in TSG-6 is particularly relevant, as one of its protective effects is its ability to suppress neutrophil influx at inflammation sites and the presentation and transport of CXCL8 through endothelial cells. By binding to glycosaminoglycans in the extracellular matrix, TSG-6 inhibits the presentation of inflammatory chemokines at the cell surface, providing a mechanism by which the cellular presentation of chemokines is regulated. Moreover, they showed that grafting activated fat in acute SCI decreases the number of CD68-positive cells at the lesion site. This suggests a reduction in macrophage invasion at the lesion site [43].

Then, it can be hypothesized that the underlying mechanisms related to both the decrease in intralesional concentration of pro-inflammatory cytokines observed and the recovery of sensorimotor functions could be ascribed to the combination of many factors such as the potential release of protective cytokines [33], the mechanic support exerted by the matrix of activated fat [43], and the presence of ADSCs in the grafted tissue [32,33]. Taken together, all these possible mechanisms could also favor endogenous neurogenesis in the damaged cord [43].

### 4.2. Activated Fat Improves Sensorimotor Recovery

Consistent with previous studies, after spinal cord contusion the animals showed an immediate significant decline in sensorimotor scores [56,57,89]. However, the BBB test, assessing interlimb coordination, revealed that the injured animals treated with activated fat exhibited consistent coordination between their forelimbs and hindlimbs during locomotion, as well as continuous weight support, contrary to injured animals treated with NaCl, which displayed uncoordinated step intervals until the end of the 12th week. Furthermore, the results of the ladder-climbing test, evaluating the integration of both sensory inputs and motor outputs, indicated that the treated animals in the AF group showed a faster recovery, starting in the 3rd week, whereas rats in the NaCl group exhibited a slower spontaneous recovery, which remained significantly lower than that of the SHAM group.

These results are in accordance with the gait analysis findings, which revealed a significantly reduced phase dispersion for the NaCl group compared to the other two groups, while no difference was observed between the SHAM and AF groups. Interestingly, this indicates a similar limb synchrony in animals of these two groups. Moreover, the achievement of a 100% regulation index between the SHAM and AF groups confirms the restoration of limb coordination during walking in animals treated with activated fat. Precise synchronization of limb movements and different body parts involved in walking is necessary for smooth and efficient locomotion. Altered coordination can lead to an unstable gait, disordered movements, and increased energy expenditure [90,91]. Thus, the improvements observed in the activated-fat-treated animals may allow for better motor control, increased stability, improved balance, and greater movement efficiency during walking. Furthermore, our results also demonstrated that the treatment with activated fat also improved the stride length, as animals belonging to this group had a longer stride length compared to rats from the NaCl group, meaning they took fewer steps to cover the same distance. This can result in smoother joint movements with a greater range of motion during walking. However, no difference was reported in the time to cross the glass plate between the NaCl and AF groups. This could be explained either by the high intra-group variability in the NaCl group for swing and stance phase times or by the fact that the speed was measured considering all the steps taken to cross the glass plate (excluding the first and last steps), whilst for the other parameters only five consecutive full steps were considered.

These results are in accordance with a previous preliminary study showing that mechanically activated human adipose tissue, when grafted in SCI affected mice, promotes neurogenesis, abrogates neuroinflammation, and favors hindlimb functional recovery one month post-injury, suggesting that fat cells may have therapeutic potential [43]. Indeed, the authors demonstrated that transplantation of activated fat at the lesion site stimulated the differentiation of endogenous ependymal stem cells into cells with a neuronal phenotype. This was evidenced by an increase in the neuronal markers β-tubulin 3 and MAP2 (microtubule-associated protein 2). Thus, the authors demonstrated the neuroprotective effects of the activated fat graft, which preserved white matter in injured and treated animals. Additionally, it allowed for increased axonal regrowth across the lesion [43].

It can be hypothesized that in our study the ADSCs present in the fat grafts played a significant role in tissue preservation and repair after SCI. The effects of ADSCs transplantation as a possible regenerative therapy for SCI has been quite extensively investigated. There are multiple studies in preclinical experimental disease models (especially rodents) in which ADSCs, alone or complexed with scaffolds or drugs, have been transplanted into the lesion site [92,93,94,95,96]. These studies have shown evidence relating to functional recovery associated with axonal regeneration [92,93,94,95]. Recently, it has been reported that ADSCs transplantation combined with training promoted neurogenesis and axonal regeneration while decreasing glial scar formation and neuronal death after spinal cord contusion in rats [96]. The feasibility of ADSCs transplantation in humans has also been explored in a phase 1 trial in which evidence regarding the safety of the cells’ intrathecal administration has been reported [97]. ADSCs have been shown to secrete various neurotrophic factors, including nerve growth factor (NGF), glial-derived neurotrophic factor (GDNF), hepatocyte growth factor (HGF), and brain-derived neurotrophic factor (BDNF) [98]. These factors modulate the PI3K/AKT pathway and the MAP kinase pathways, which are involved in neuronal differentiation, survival, and axonal regrowth [99]. Furthermore, endothelial progenitor cells found in adipose tissue secrete vascular endothelial growth factor (VEGF). This factor stimulates angiogenesis, supplying the injured tissue with oxygen and nutrients necessary for tissue repair [100]. Additionally, VEGF modulates the MAPK/Erk1/2 pathway, which is involved in neuronal survival, cell differentiation, and axonal regrowth [101].

Other studies also reported that adiponectin, a cytokine released by adipose tissue, was able to promote neurogenesis and synaptic plasticity through its action on AdipoR1 and AdipoR2 receptors [102,103,104]. Furthermore, adiponectin exerts beneficial actions on endothelial homeostasis, acting as a regulator of endothelial nitric oxide synthase (eNOS), a key determinant of endothelial function and angiogenesis [105]. It has been demonstrated that cerebral ischemia leads to a reduction in lesion size and allows for functional recovery due to its strong angiogenic potential [106,107].

Additionally, leptin, a potent hormone released by adipocytes, showed neuroprotective effects, and the ability to improve sensorimotor recovery after SCI [87,108]. Moreover, a more recent study demonstrated the neuroprotective capacity of leptin via the activation of the JAK2/Stat3 pathway, which increases the expression of Caveolin-1 in mice after SCI. This protein plays a critical role in promoting neuronal survival, reducing excitotoxicity following injury [109].

Therefore, we can hypothesize that the grafting of activated fat, due to the combined actions of ADSCs, endothelial cell progenitors, and the released adipokines, may limit the extension of secondary injury. Moreover, it could shape the injured microenvironment, reduce glial scar formation, promote neurogenesis, facilitate axonal regeneration, allow the formation of new functional synapses between host neurons and neurons formed after the graft, and revascularize the injured tissue, thereby replenishing it with the necessary nutrients and oxygen for proper functioning [33,39,43,110,111,112,113,114]. Thus, it appears that the therapeutic effects on sensorimotor recovery result from the synergistic action of cells contained in the activated fat and trophic factors released.

Our study highlights the potential of activated fat to promote sensorimotor recovery after acute SCI in rats, paving the way for the potential restoration of sensorimotor loop functions.

### 4.3. Activated Fat Restores Sensorimotor Loops and Their Supraspinal Command

SCI affecting the transmission of information through the ascending and descending spinal pathways may disrupt the functioning of the sub-lesional sensorimotor loops, leading to disorders including spasticity, muscle paralysis, loss of reflexes, and/or hypotonia [115].

In healthy animals, a 3 min muscle stimulation at 10 Hz typically induces muscle fatigue, activating metabosensitive muscle afferent pathways and resulting in an increase in ventilation frequency and amplitude [116]. These ventilatory adjustments to metabosensitive afferent fibers’ activation with repetitive stimulation of the Tibialis anterior muscle performed under regional circulatory occlusion allows the isolation and maintenance of the neural drive while eliminating hormonal communication [116]. The ventilatory adjustments rely on the functional integrity of the neuronal network allowing the transmission of impulses from the ascending pathway to the ponto-bulbar respiratory centers. After SCI, the ventilatory response to muscle stimulation is diminished or abolished, depending on the size of the lesion and the spared ascending pathways [61,68,69,71,117]. Our results confirm these conclusions. SCI animals receiving NaCl showed a lack of ventilatory response. In contrast, the AF group demonstrated a similar response to the SHAM group, indicating the beneficial effect of activated fat in re-establishing connections with the ponto-bulbar ventilatory centers.

The H-reflex is a neurophysiological phenomenon involving neural circuits of the monosynaptic stretch reflex of a muscle. It is triggered by the stimulation of Ia afferents, causing excitation of the α-motoneurons and subsequent contraction of the muscle without the intervention of supraspinal centers. The literature reports that high-frequency nerve stimulation leads to an increase in the H-reflex amplitude due to the lack of inhibitory feedback from supraspinal pathways in animals with SCI [68,69,117,118,119]. Indeed, the interruption of the descending pathway and disorganization of spinal networks lead to spasticity, a condition in which muscles stiffen or tighten, preventing normal fluid movement. The muscles remain contracted, and resist being stretched, thus affecting movement. Hyperreflexia is the most studied component of spasticity that may result from, among other things, a decrease in presynaptic inhibition of Ia afferents by supraspinal structures, changes in α-motoneuron excitability (persistent inward current), and/or changes in synaptic transmission (release of neurotransmitter, post-synaptic receptor, number of synapses, …) in the reflex arc [120,121,122]. Thus, to evaluate this component of spasticity, we recorded the H-reflex and measured the Hmax/Mmax ratio under different frequencies of stimulation. In the absence of SCI, the Hmax/Mmax ratio decreases when the frequency of stimulation increases. However, after an SCI, an attenuation, or indeed the abolition, of the RDD [123] has been identified. The study of the H-reflex amplitude in response to various nerve stimulations can serve as a valuable indicator of neural plasticity and/or spinal tract regeneration, providing information on the efficacy of therapeutic interventions for recovery after an SCI [56,66].

In our study, the electrophysiological recordings performed twelve weeks after SCI showed that AF-treated animals exhibited a significant post-activation depression of the H-reflex, as in non-injured animals, whereas no variation was observed in the NaCl group, despite an increase in electrical stimulation frequency. Additionally, we observed a restoration of ventilatory adjustments in response to electrically induced fatigue of the Tibialis anterior muscle’s metabosensitive fibers in animals grafted with activated fat, unlike those in the NaCl group. These results strongly underline the beneficial effect of activated fat on the restoration of sensorimotor loop functionality and the re-establishment of communication in ascending and descending pathways, which may be responsible for improved motor skills and posture in treated animals. Moreover, this also suggests that activated fat enabled a reorganization and/or regeneration and functionalization of the spinal neuronal network.

Recently, Yuan et al. demonstrated significant recovery of motor-evoked potentials recorded at the spinal cord level in rats with SCI infused with ADSCs at the lesion level [124]. Furthermore, another study indicated that the administration of ADSCs in combination with a fibrin matrix in injured rats resulted in the restoration of post-activation depression of the H-reflex 60 days after the injury [125]. Keikhaei et al. also showed a post-activation depression of the H-reflex in animals transplanted with ADSCs twelve weeks after spinal cord contusion [96] and other studies have demonstrated the neuroprotective effects of adiponectin on neural circuits [126,127,128,129]. Finally, certain studies have concluded that the RDD restoration is correlated to an increase in neurotrophic factor proteins (BDNF, NT-3, and NT-4) [130,131,132,133] and an increase in BDNF upregulating KCC2 expression [134]. Because ADSCs contained in the fat have been shown to secrete NGF, GDNF, HGF, and BDNF, we cannot exclude an action of this neurotrophic factor on the restoration of the RDD in animals receiving the activated fat.

In view of this scientific literature, our results suggest that the restoration of the sensorimotor loop and spinal pathways after SCI could be attributed to the combination of different effects induced by activated fat, as already described above.

## 5. Conclusions

The current study takes place in the critical clinical context of acute SCI, a major challenge with a significant impact on patients’ quality of life. Our aim was to provide innovative solutions to this problem by exploring the potential of adipose tissue, recognized for its high regenerative potential and accessibility, thereby potentially minimizing discomfort for patients. The particularity of our approach lies in the prior activation of adipose tissue, an innovation compared with previous studies. This specific activation is crucial for its impact on acute anti-inflammatory mechanisms and its long-term stimulation of tissue regeneration. It is essential to underline that adipose tissue is not classified as a heterologous treatment due to its minimal manipulation. In accordance with the criteria of the International Society of Stem Cell Research (ISCCR) and regulatory agencies such as the Food and Drug Administration and the European Medicines Agency, this approach does not exceed the limits to be considered a cell therapy [135,136,137]. In addition to its positive regulatory status, the results of this study show the beneficial effects of using human mechanically activated fat in acute SCI due to its ability to (i) reduce lesion-induced neuroinflammation, (ii) restore spinal communication between supra- and sub-lesional spinal cord, and (iii) improve sensorimotor recovery.

In conclusion, this study shows that the use of activated fat for acute SCI represents a potential compelling therapeutic strategy. However, many aspects need deeper investigation, particularly with regards to the underlying mechanisms behind the observed recoveries and the importance of each element contained in the fat. Additional studies, including detailed histology analyses, would provide further insights into the underlying mechanisms behind the restoration of segmental neuronal network functions and the transmission of information from these networks to supraspinal centers.

## Figures and Tables

**Figure 1 cells-13-00182-f001:**
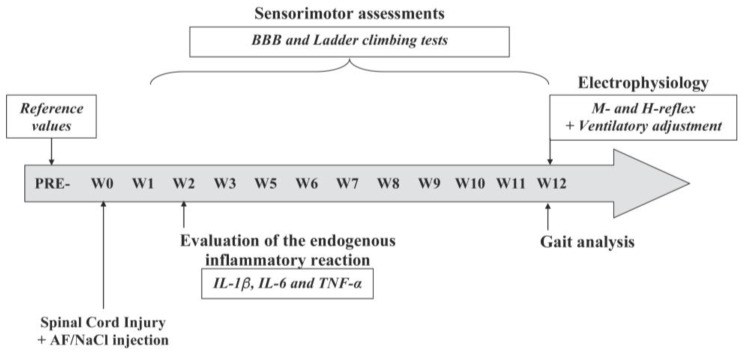
Experimental protocol. One week before the surgery, baseline values (PRE-) for each functional test were recorded. The surgery and the application of saline and activated fat were performed at W0. Subsequently, sensorimotor recovery was measured once a week for 12 weeks (from W1 to W12) using behavioral tests (BBB and ladder-climbing tests). The endogenous inflammatory response was analyzed at W2 (2 weeks after the injury). At the end of W2, a gait analysis test was performed, and electrophysiological tests recorded the M-wave and the H-reflex.

**Figure 2 cells-13-00182-f002:**
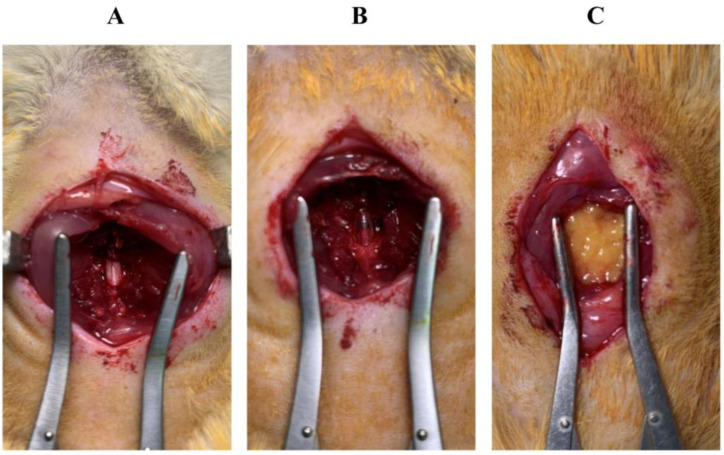
Exposed spinal cord. (**A**) Before the lesion. (**B**) After the contusion. (**C**) After transplantation. The contused spinal cord was covered with activated fat before the wounds were closed.

**Figure 3 cells-13-00182-f003:**
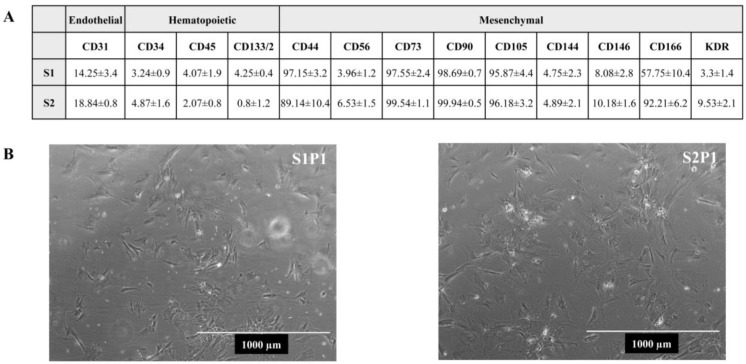
In vitro characterization of human adipose tissue cell cultures from AF. (**A**) Expression of cell surface markers. All percentages were obtained by flow cytometry analysis [27]. Results are expressed as mean ± SD of three independent experiments. (**B**) Live morphology of human adipose tissue-derived mesenchymal stem cells (hADSCs) obtained from AF. Pictures refer to two different preparations (sample 1: S1, and sample 2: S2) kept in culture at passage 1 (P1).

**Figure 4 cells-13-00182-f004:**
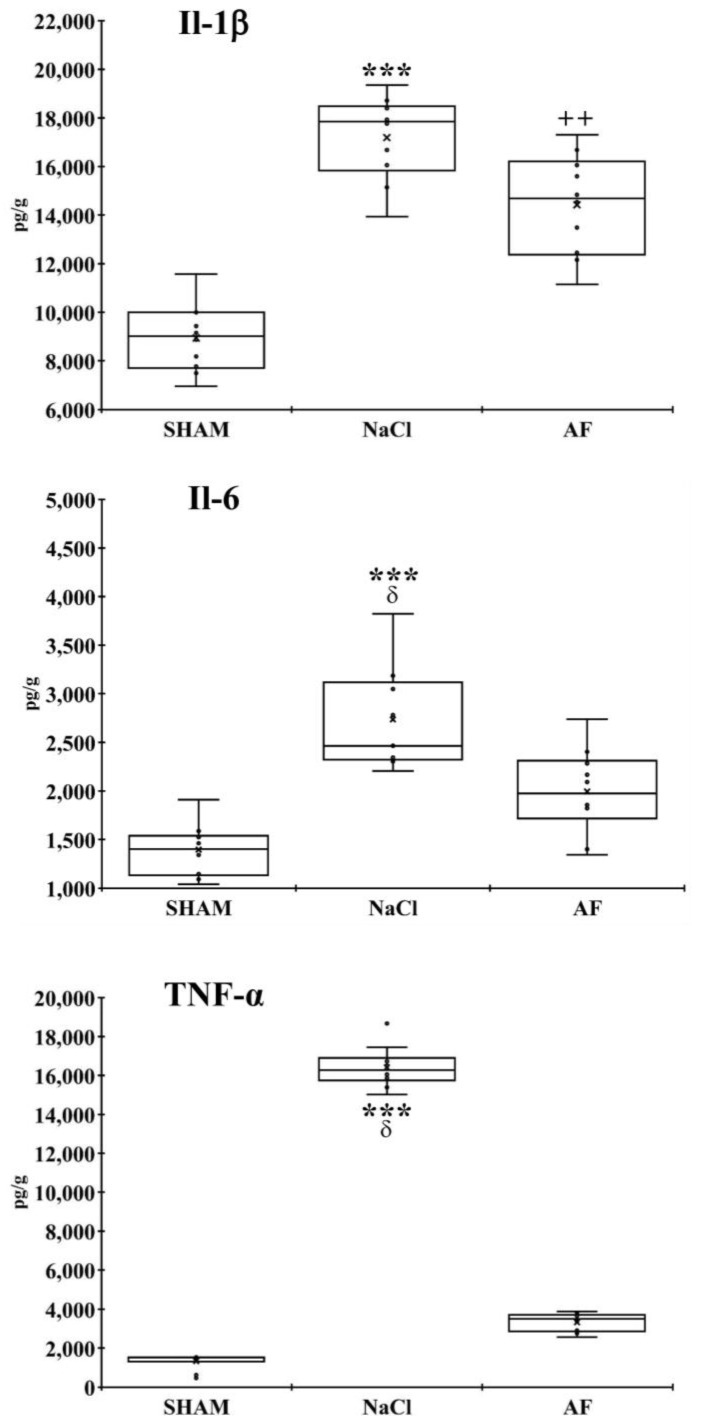
Inflammatory reaction at the lesion site. Concentrations of IL-1β, IL-6, and TNF-α at the site of injury in the SHAM, NaCl, and AF groups, two weeks post-injury indicated that the level of inflammation was higher in the NaCl group compared to the two other groups. Measurements were performed twice. Significant difference is indicated by * (NaCl group vs. SHAM group), + (AF group vs. SHAM group), or δ (NaCl group vs. AF group); 1 symbol, *p* < 0.05; 2 symbols, *p* < 0.01; and 3 symbols, *p* < 0.001. x and

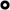
 indicate the mean and individual response, respectively. n = 5 per group.

**Figure 5 cells-13-00182-f005:**
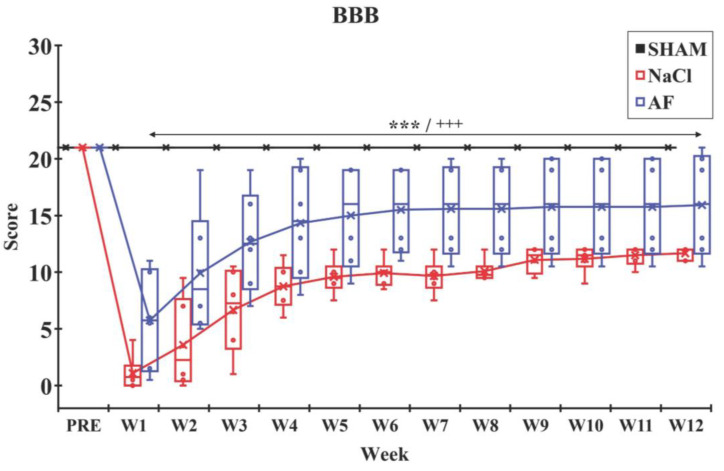
BBB locomotor rating scale. After the SCI, the BBB score in each injured group dropped significantly, and then a recovery was observed until W12. Data indicate a faster recovery in the AF group compared to the NaCl group. Significant difference in the BBB scores is indicated by * (NaCl group vs. SHAM group) or + (AF group vs. SHAM group); 3 symbols, *p* < 0.001. x and 
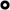
 indicate the mean and individual response, respectively. n = 6 per group.

**Figure 6 cells-13-00182-f006:**
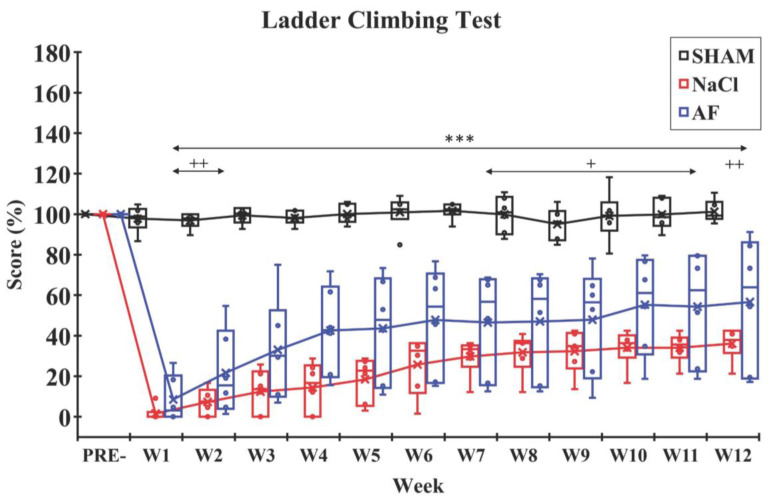
Ladder-climbing test. After the SCI, the climbing score in each injured group dropped significantly, and then a slow recovery was observed until W12. Significant difference in the climbing scores is indicated by * (NaCl group vs. SHAM group) or + (AF group vs. SHAM group); 1 symbol, *p* < 0.05; 2 symbols, *p* < 0.01; and 3 symbols, *p* < 0.001. x and 
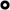
 indicate the mean and individual response, respectively. n = 6 per group.

**Figure 7 cells-13-00182-f007:**
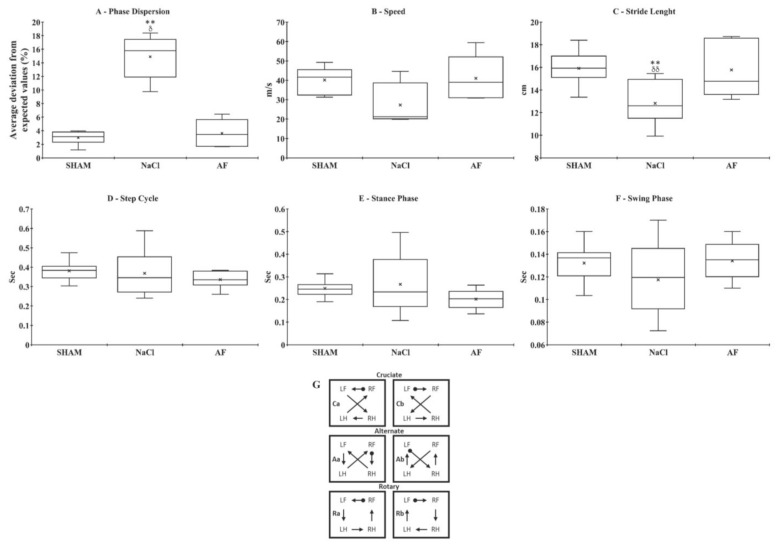
Gait analysis parameters. (**A**) Phase dispersion. Average deviation between hindlimbs from the expected value of 50%: animals treated with NaCl showed a significant increase compared to the SHAM and AF groups. (**B**) Average speed. The average crossing speed on the glass plate was similar in the 3 experimental groups. (**C**) Stride length. Average stride length between hindlimbs was significantly decreased in the NaCl group compared to the SHAM and AF groups. (**D**) Step cycle. Average step cycle between hindlimbs was similar in the 3 experimental groups. (**E**) Stance phase. Average stance phase between hindlimbs was similar in the 3 experimental groups. (**F**) Swing phase. The average time of the swing phase between hindlimbs was similar in the 3 experimental groups. (**G**) Step sequence patterns seen in normal rats (adapted from [20]). Significant difference is indicated by * (NaCl group vs. SHAM group) or δ (NaCl group vs. AF group); 1 symbol, *p* < 0.05; and 2 symbols, *p* < 0.01. x indicates the mean. n = 6 per group.

**Figure 8 cells-13-00182-f008:**
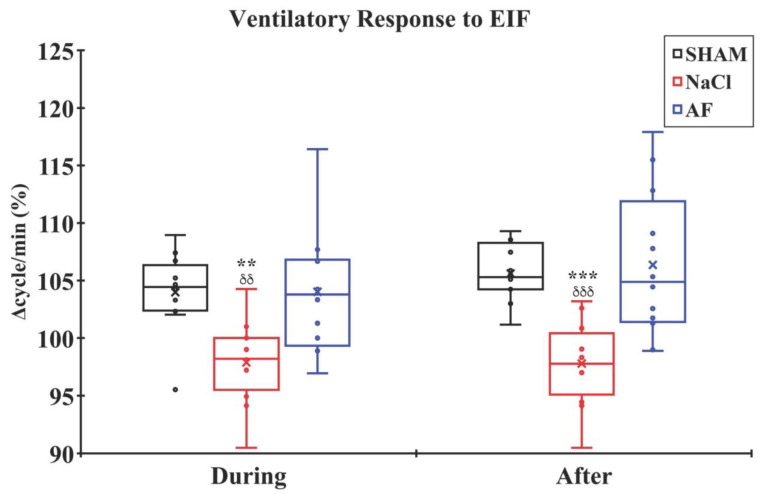
Ventilatory adjustment to 3 min EIF. The ventilatory activity significantly increased in the SHAM and AF groups during and after 3 min muscle stimulation of the Tibialis anterior muscle under local regional circulatory occlusion. Significant difference in the mean ventilatory frequency is indicated by * (NaCl group vs. SHAM group) or δ (NaCl group vs. AF group); 2 symbols, *p* < 0.01; and 3 symbols, *p* < 0.001. x and 
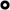
 indicate the mean and individual response, respectively. n = 6 per group.

**Figure 9 cells-13-00182-f009:**
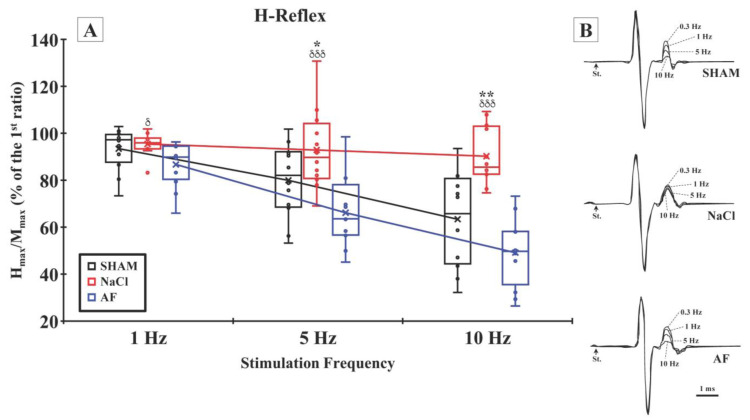
H-reflex rate sensitivity. (**A**) Hmax/Mmax ratio, measured after increasing the frequency of 548 stimulation, showed, from 1 Hz stimulation, a significant depression in the SHAM and AF groups. For a given frequency, significant difference in the Hmax/Mmax ratio is indicated by * (NaCl group vs. SHAM group) or δ (NaCl group vs. AF group); 1 symbol, *p* < 0.05; 2 symbols, *p* < 0.01; and 3 symbols, *p* < 0.001. x and 
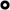
 indicate the mean and individual response, respectively. n = 6 per group. (**B**) Representative examples of H-reflex recorded at different stimulation frequencies in the SHAM (upper traces), NaCl (middle traces), and AF (bottom traces) groups.

## Data Availability

Data will be available upon request from the authors.

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
