# Peer review of "Activated Human Adipose Tissue Transplantation Promotes Sensorimotor Recovery after Acute Spinal Cord Contusion in Rats"

_cells, 2024, doi:10.3390/cells13020182_

Round 1

Reviewer 1 Report

Comments and Suggestions for Authors

Manuscript ID: cells-2763209

Activated Human Adipose Tissue Transplantation Promotes Sensorimotor Recovery After Spinal Cord Contusion in Rats by Bonnet and colleagues

The authors have applied mechanically activated human adipose tissue to acute, severe contusion injuries of the adult rat and have used a number of functional/behavioural and electrophysiological tests to demonstrate some degree of repair. The manuscript is generally well written but there is a moderate number of grammatical/linguistic errors that should be corrected/modified to improve it, such as in line 64: “ cell dead replacement”  could be “replacement of dead cells”. Also in line 98:  “ spinal cord in a consolidated rat experimental model of SCI” could be “in a reproducible and widely used used rat experimental model”. Lines 529-531: “These events, which are involved in healing processes, can induce, if persist, tissue degeneration, further exacerbating the damage caused by the injury and limiting spontaneous regeneration [6]”. The authors have simply not chosen the best terms in these (and numerous other) cases - this issue could easily be resolved if the publishers perform their own linguistic checks or if the authors request the paper be checked by a competent native English speaker.

Major comments:

My biggest concern about the manuscript is the total lack of any supporting morphological data which leaves the authors and the reader in the dark about the mechanism(s) that bring about the beneficial effect. Has the apparent anti-inflammatory effect of the implanted adipose tissue resulted in quantifiable sparing of spinal cord white matter? This would have been simple for the authors to address  - they even mention in section 2.9, line 369 that “spinal cord was removed to verify the extent of the lesion“ but failed to describe methodologically how the tissue was processed and analysed, and what relevant data was generated from this tissue. The authors also stated in lines 519-521 that “This finding suggests that mechanically activated fat may provide neuroprotective and/or neuroregenerative factors that can increase tissue preservation following the secondary injury created by a series of biological and functional changes and/or regrowth of injured axons” – if the authors presented the relevant data on lesion size, they would be able to make a firm conclusion regarding neuroprotection.

 It would also be important to know the morphological status of the adipose tissue just prior to implantation, at the 2 week observation point mentioned and at the 12 week end-point. Does the implanted tissue (or the donor ADSC) survive or does it degenerate? Is there any immune indication of rejection of the implant? None of these issue have been addressed.

The authors report the 200Kdyn contusion injury as being “severe” – this is certainly true for the smaller rats (200-225g) used in the supporting citation (47), but is this also the case for the bigger (250-300g) rats used in the present study? I personally think that it is likely, but don’t think that scientific papers should simply assume that this is the case, bearing in mind that the animals used in the present study are between 10-33% bigger. I believe that is reasonable and proper for the authors should back up such comments with their own hard data.

Lines 562-563 the athors state “In our present study, the contusion model used does not induce a breach of the blood-spinal cord barrier [41,47].” This is wrong – such large injuries destroy the cells at the lesion site, including the astrocytes, pericytes and endothelial cells  – resulting in massive bleeding. I think that this constitutes a breach of the blood-spinal cord barrier. Furthermore, citations 41 and 47 may be inappropriate since neither mention the blood-spinal cord barrier.

Although the authors nicely show the BBB and ladder climbing data, they fail to state clearly that there is no significant difference between the performance of the NaCl and AF groups.  In the figure legend for figure 4 there is a comment by the authors that there is a Significant difference in the climbing scores is indicated by a * (NaCl group vs. SHAM group), + (AF group vs. SHAM group)  and sigma symbol (NaCl group vs. AF group). There is, however, no sigma symbol in the figure.

Lines 161-165 the authors state “ To activate the lipoaspirated adipose tissue, a method described in previously published research was followed [19,28]. Briefly, 10-15 ml of lipoaspirated adipose tissue was mechanically activated by orbital shaking at a speed of 97 x g (N.kg-1) for a period of 7 minutes at room temperature. The fat was activated 30-60 min before transplantation. The resulting product was referred to as mechanically activated lipoaspirate or activated fat (AF).” However, the authors make no attempt to demonstrate that the cells that they are working with have actually become “activated” – this should be done.

Minor comments:

What is the reason for the immediate injection of NaCl on the lesion site ? Was this also performed for the AF group?   

Line 223-224 states: Following SCI, animals in the NaCl and AF groups received a NaCl and AF injectionrespectively, in the epicenter of the lesion (Supplemental Figure S1)”.The figure legend for figure 1 states: “The surgery and injections of NaCl and AF were performed at W0” however supplementary figure 1S C doesn’t appear to show AF that has been injected. There is confusion as to whether injections were on or in the epicentre of the lesion – this must be carefully corrected.

How much NaCl and AF were applied to the lesions? What was the reason for the choice of these (unknown) amounts?

Lines 167-169: the authors state “To demonstrate that the procedure of activation of lipospirated adipose tissue do not damage the tissue, we isolated human adipose tissue cells using a procedure previously 168 described [19]. “ The fact that cells can migrate out of the 1cm3 fat sample onto tissue culture plastic and subsequently undergo expansion doesn’t prove that the activation process did not damage the tissue. Did the authors compare cell numbers migrating from activated and non-activated fat?              

Comments on the Quality of English Language

The manuscript is generally well written but there is a moderate number of grammatical/linguistic errors that should be corrected/modified to improve it, such as in line 64: “ cell dead replacement”  could be “replacement of dead cells”. Also in line 98:  “ spinal cord in a consolidated rat experimental model of SCI” could be “in a reproducible and widely used used rat experimental model”. Lines 529-531: “These events, which are involved in healing processes, can induce, if persist, tissue degeneration, further exacerbating the damage caused by the injury and limiting spontaneous regeneration [6]”. The authors have simply not chosen the best terms in these (and numerous other) cases - this issue could easily be resolved if the publishers perform their own linguistic checks or if the authors request the paper be checked by a competent native English speaker.

Author Response

Dear author, we attached the word file including the point by point response.

Reviewer 2 Report

Comments and Suggestions for Authors

The authors examined whether transplanted activated human fat (AF) could aide in restoration of function following a spinal cord injury (contusion model).  The rationale being that adipose tissue also contain protective adipokines such as leptin, angiotensinogen, resistin etc., as well as anti-inflammatory compounds.  They demonstrate that AF can reduce neuroinflammation at 2 -week post injury and aide in sensory motor recovery over a 12 week period assessed by BBB score, ladder walking, gait analysis and m/H reflex FDD.  The results are very promising, with methods well detailed and would be of interest to a large audience.  However, I have several concerns, a few major but mostly minor that should be addressed. 

Overall methods section is very long and redundant in many sections.  Please consider reducing length. For instance lines 353-360  and 363-365 does not belong here, could be in the  results or discussion. 

Line 215 and 219: please indicate whether T10  refers to the T10 laminae or the T10 spinal segment.

Line 234: states 5mm rostral and 5mm caudal were collected. Why was such a large area collected and homogenized if the hypothesis was that AF was reducing inflammation at the site of injury. 

Line 306: What is the rationale for 5 best steps?  Is this 5 best consecutive steps.  Was 5 steps chosen as that was the maximum number of steps these rat could make .

Line 345: frequency dependent depression is referred to as FDD and not FDR.

For all figures: please include n for each group in the figure legends and as these are low numbers it would be nice to plot the individual responses in the figures on top of the box plots.  This will demonstrate the variability in the AF as based on at least fig 3 and 4 there is large variability between responders and potential non-responders in the AF group. 

Figure 6: would benefit by including the actual reflex responses for each group and for each frequency.  This would people not within this field understand what is happening or decreasing. 

Discussion: based on the novelty of using AF, it would be beneficial for the reviewers to speculate or provide relevant literature of the potential mechanism of how AF is impacting the epicentre of the spinal contusion to lead to improvement.  It was touched on briefly that adiponectin could cross the blood-brain-barrier.  What about the other molecules discussed in the intro, can these readily cross the blood brain barrier.  I assume with the contusing the epidural layer remains intact, so it would be essential to describe how this is occuring.

Major concern is the lack of detail in the methods regarding AF injection (223-227).  Authors state that AF was injected into the epicenter of the lesion.  If so please provide details about the volume of injection, the method of injection, period of injection time.  When looking at supplemental figure 2, it appears that AF is actually just placed on top of the exposed duramater of the spinal cord. 

If that is the case, how are the authors certain that the AF remained in place at the contusion site after surgery and during recovery.  With out this information, it is difficult to assess whether AF is beneficial at the site of injury.  Was there any follow-up to see if AF remained at the 2 week period in which tissues were collected. Or any assessment that the factors penetrated to the site of injury.  Similar was the NaCL actually injected into the epicentre. If so, that would suggest the epidural was punctured in the NaCL group and potentially not in the AF group based on what is seen in supplemental figure 2

Similar, should a different control have been used instead of the Nacl injection that would mimic the weight or voume of the transplanted AF on top of the spinal cord.  Once cord speculate that having a mass sit on top of the injured spinal cord in itself could reduce inflammation by mechanically pushing the fluid and inflammation away from the site of injury. 

Ideally, the impact of this paper could greatly be increased if the mechanisms of how it is beneficial were also explained or examined and confirmation that indeed the AF remained intact and viable at the site of injury.

Comments on the Quality of English Language

n/a

Author Response

(The authors gave the same response as above.)

Reviewer 3 Report

Comments and Suggestions for Authors

This is a very interesting study concerning the animal spinal cord injury model treated by activated lipoaspirated adipose tissue from the human. The author concluded that tissue transplantation modulated the inflammatory response contributed to the improvement in neurological outcome. But there were some basic debates that the authors should make a point- point response.

First, in tissue transplantation from human to rats, there should be a strong tissue rejection. But there were no any immunosuppression agents used in this study. The authors should describe this point clearly why there was no strong tissue rejection.  

Second, the experimental data was not very convincing. The authors at least should present the immunohistology alteration in the injured spinal cord including the typical markers of neuron, astrocyte, or microglia as well as fiber tract or the number of synapses.

Third, the tissue transplantation included the mixed cells. What is the major function in this study ? either adipocyte, endothelia cells or stem cells. The authors at least should trace these cells changes in the injured cord after the transplantation.

Comments on the Quality of English Language

The English editing is very nice. 

Author Response

(The authors gave the same response as above.)

Reviewer 4 Report

Comments and Suggestions for Authors

1.Is not clear if this work is for acute or chronic patients.

2.Must be clear how we deal with the scar.

3.Ethical issues regarding human adipose to animals.

4.I'm doing human autologous olfactory ensheathing cells transplantation, and the work is in Fase II on humans(therefor i answered if i have any potential conflict of interest).https://www.researchgate.net/publication/358525072_Clinical_results_in_treatment_of_patients_with_spinal_cord_injuries_by_transplantation_of_olfactory_mucosa_autografts_a_case-series?_tp=eyJjb250ZXh0Ijp7ImZpcnN0UGFnZSI6ImhvbWUiLCJwYWdlIjoicHJvZmlsZSIsInByZXZpb3VzUGFnZSI6ImhvbWUiLCJwb3NpdGlvbiI6InBhZ2VDb250ZW50In19

Above all, this work was very well presented and if could give answeres in my queries, would touch the maximum

Author Response

(The authors gave the same response as above.)

Reviewer 5 Report

Comments and Suggestions for Authors

In order to improve the manuscript, several changes should be included in the final version:

In the introduction, the authors should include more epidemiological data on spinal cord injury, especially the incidence on motor alterations and sensory alterations after spinal cord injury. Likewise, they must also specify more what type of sensory alteration is the most common after a spinal cord injury, that is, whether it is a mechanoreceptive, thermoreceptive or nociceptive sensory dysfunction. Please include all this information in the final version of the manuscript.

What is the direct and indirect economic cost of a patient with spinal cord injury? Please include this information in the final version of the manuscript.

Which of the pathophysiological changes of spinal cord injury contribute to motor dysfunctions and which to sensory dysfunctions? Please include this information in the final version of the manuscript.

The latest research in patients with spinal cord injury demonstrates that spinal neurostimulation allows some patients to regain the ability to move. Likewise, transplantation of olfactory ensheathing glia cells has had motor success in partial spinal cord lesions. Despite this, certain treatments remain ineffective for motor and sensory recovery after spinal cord injury. Authors should include this information in the introduction section of the manuscript, and specify which treatments remain ineffective.

The use of transplants of stem cells from adipose tissue, or even transdifferentiated cells from adipose tissue, have already been tested in experimental models of spinal cord injury. In this context, the authors must highlight the novelty of the present study compared to similar works published previously.

In the methodology section it is not clear how the cells are administered to the spinal cord. Have the cells been injected into the parenchyma of the injured spinal cord? It is necessary to describe in detail how the adipose tissue cells have been implanted in the spinal cord of the rats that have received the contusion. What concentration of cells have been implanted? What is the phenotype of implanted cells? What chemical medium was used to implant the cells? Has it been saline (NaCl)? It is not very common to use NaCl as a means of survival of the cells that must be implanted. The usual thing is to use a medium such as DMEM, MEM, etc. Please include all this relevant information in the final version of the manuscript.

The authors must include quantitative data on the variation in body weight of the animals in the different groups throughout the follow-up.

The authors must include electrophysiological recordings from the different experimental groups.

What is the degree of cellular rejection of the xenotransplantation performed by the authors in this manuscript? Please include information about this in the results section.

The authors should discuss the mechanism by which mechanically preactivated adipose cells reduce inflammation underlying spinal cord contusion. This point is very relevant and should be discussed in depth. Please include this information in the final version of the manuscript.

What the authors propose in lines 620-627 of the discussion is very interesting. However, they must include evidence demonstrating this hypothesis. Bibliographic citations must be included in this paragraph of the discussion, demonstrating that the authors' hypothesis has a scientific basis. Please include this information in the final version of the manuscript.

The authors should discuss the H reflex further and in greater depth. What does this H reflex represent? How does this H reflex change in a subject with spinal cord injury? What does the recovery of the H reflex represent? Authors should include this information in the final version of the manuscript.

Is sensorimotor recovery more related to the recovery of gait-locomotion or to the recovery of the H reflex? or with both parameters? Which of the two parameters has the most influence on better sensory-motor recovery? How do inputs from the spinal descending pathways influence sensorimotor recovery? What spinal circuits are involved in sensorimotor recovery? Has the implantation of adipose tissue cells improved the function of these circuits and does this translate into better sensorimotor recovery? The authors should answer these questions and discuss them in depth in the discussion section of the manuscript.

What is the clinical translation of the pre-clinical research carried out by the authors in this manuscript?

Figures included in supplementary material should be moved to the body of the manuscript, and not as supplementary material. They are very relevant images, which must appear in the manuscript.

Author Response

(The authors gave the same response as above.)

Round 2

Reviewer 1 Report

Comments and Suggestions for Authors

The authors have answered all the comments that were raised.

However, their answer to comment #3 needs to be modified: 

Auithors response P16 L570-: “Several studies have investigated the use of allogeneic transplants of human ADSCs to repair the spinal cord after SCI [40,73–75]."

Grafting of human donor tissue to rat or mouse host animals is not allogeneic, but rather xengeneic

Comments on the Quality of English Language

There are still several obvious minor errors in the quality of the language used, but they do not prevent the reader from understanding the authors' meaning.   

Author Response

Dear reviewer, please find reponse in the attached file.

Reviewer 2 Report

Comments and Suggestions for Authors

The authors have made extensive revisions to the methods, results, and discussion and have satisfied my previous concerns. 

Comments on the Quality of English Language

n/a

Author Response

(The authors gave the same response as above.)

Reviewer 3 Report

Comments and Suggestions for Authors

The authors tried to make a point-point response.  The manuscript should be published.   

Comments on the Quality of English Language

This manuscript need mild English editing. 

Author Response

(The authors gave the same response as above.)

Reviewer 5 Report

Comments and Suggestions for Authors

The authors have responded to most of the reviewer's suggestions, and this has allowed the quality of the manuscript to significantly increase.

Author Response

(The authors gave the same response as above.)
